# Yin and Yang of NADPH Oxidases in Myocardial Ischemia-Reperfusion

**DOI:** 10.3390/antiox11061069

**Published:** 2022-05-27

**Authors:** Shouji Matsushima, Junichi Sadoshima

**Affiliations:** 1Department of Cardiovascular Medicine, Faculty of Medical Sciences, Kyushu University, Fukuoka 812-8582, Japan; matsushima.shoji.056@m.kyushu-u.ac.jp; 2Department of Cell Biology and Molecular Medicine, Rutgers New Jersey Medical School, Cardiovascular Research Institute, Newark, NJ 07103, USA

**Keywords:** NADPH oxidase, oxidative stress, mitochondria, endoplasmic reticulum, energy metabolism

## Abstract

Oxidative stress is critically involved in the pathophysiology of myocardial ischemic-reperfusion (I/R) injury. NADPH oxidase (Nox) 2 and 4, major sources of reactive oxygen species (ROS) in cardiomyocytes, are upregulated in response to I/R. Suppression of Nox-derived ROS prevents mitochondrial dysfunction and endoplasmic reticulum (ER) stress, leading to attenuation of myocardial I/R injury. However, minimal levels of ROS by either Nox2 or Nox4 are required for energy metabolism during I/R in the heart, preserving hypoxia-inducible factor-1α (HIF-1α) and peroxisome proliferator-activated receptor-α (PPARα) levels. Furthermore, extreme suppression of Nox activity induces reductive stress, leading to paradoxical increases in ROS levels. Nox4 has distinct roles in organelles such as mitochondria, ER, and ER-mitochondria contact sites (MAMs). Mitochondrial Nox4 exerts a detrimental effect, causing ROS-induced mitochondrial dysfunction during I/R, whereas Nox4 in the ER and MAMs is potentially protective against I/R injury through regulation of autophagy and MAM function, respectively. Although Nox isoforms are potential therapeutic targets for I/R injury, to maximize the effect of intervention, it is likely important to optimize the ROS level and selectively inhibit Nox4 in mitochondria. Here, we discuss the ‘Yin and Yang’ functions of Nox isoforms during myocardial I/R.

## 1. Introduction

Coronary heart disease (CHD) is a major cause of death and disability worldwide. Reperfusion therapy by coronary revascularization is one of the most well-established treatments to reduce myocardial injury and limit the size of myocardial infarction resulting from acute ischemic heart disease. However, reperfusion therapy is accompanied by myocardial ischemia-reperfusion (I/R) injury, which itself can induce cardiomyocyte death and myocardial infarction, eventually causing cardiac remodeling and dysfunction long-term. Although myocardial I/R injury is intimately involved and directly related to poor outcomes in CHD patients, there is still no effective therapy [1]. Nicotinamide adenine dinucleotide phosphate (NADPH) oxidases (Noxs), a family of enzymes dedicated to the production of reactive oxygen species (ROS), are intimately involved in the pathophysiology of myocardial I/R injury. In this review, we summarize the physiological and pathological functions of Nox2 and Nox4, two major isoforms in the heart, during myocardial I/R injury and discuss their Yin and Yang aspects.

## 2. Oxidative Stress and Myocardial I/R Injury

### 2.1. Roles of Oxidative Stress in Myocardial I/R Injury

Acute myocardial ischemia causes a marked reduction in the tissue oxygen content and switches cell metabolism from aerobic to anaerobic respiration, resulting in energy depletion, lactate production, and a decrease in intracellular pH. The acidic condition inhibits the opening of the mitochondrial permeability transition pore (mPTP) and alters ion homeostasis, leading to cardiomyocyte death and cardiac dysfunction [2]. In the presence of residual oxygen or during low-flow ischemia, ROS are produced in the myocardium [3].

On the other hand, during reperfusion, reoxygenation in the ischemic lesion induces robust production of ROS, including hydrogen peroxide (H_2_O_2_), superoxide radical anion (O_2_^−^), hydroxyl radical (OH•), and peroxynitrite anion (ONOO^−^) [4,5,6]. Oxidative stress, defined as excess production of ROS relative to the antioxidant defense, critically mediates myocardial I/R injury by causing oxidative damage to DNA, lipid, and protein [7]. The enhancement of mPTP opening and sarcoplasmic reticulum (SR) calcium loading facilitates ROS-induced myocardial I/R injury [8].

### 2.2. Sources of Oxidative StressiIn Myocardial I/R Injury

There are several sources of ROS during myocardial I/R injury, including the mitochondrial electron transport chain (ETC) [9,10,11], xanthine oxidase [12,13], and Noxs [2,14,15]. Of these, Noxs are major contributors to ROS production during myocardial I/R injury [16]. Noxs are transmembrane enzymes that produce O_2_^−^ and H_2_O_2_ from molecular oxygen, using NADPH or NADH as an electron donor [17]. Noxs are expressed in many cell types, including cardiomyocytes, endothelial cells, smooth muscle cells, and leukocytes [18,19]. To date, seven isoforms of the Nox family proteins, namely Nox1-Nox5, Duox1, and Duox2 (recently also termed Nox6 and Nox7, respectively), have been identified. Nox1, Nox3 and Nox5 produce superoxide, whereas Duox1, Duox2, and Nox4 largely produce hydrogen peroxide [19,20]. Nox1 is expressed in endothelial cells, vascular smooth muscle cells (VSMCs), fibroblasts, and cardiomyocytes [21]. Nox2 was first identified as gp91*^phox^* in neutrophils [17]. Nox4 was originally identified as a renal-specific oxidase [22]. Nox2 and Nox4 are widely expressed in endothelial cells, VSMCs, fibroblasts, and cardiomyocytes [23,24]. Nox1, Nox2, and Nox4 are major isoforms involved in the pathophysiology of cardiovascular diseases, including myocardial I/R injury [14].

## 3. Pathophysiological Roles, Regulation, and Subcellular Localization of Nox Isoforms in the Heart

The pathophysiological role of endogenous Nox isoforms in the heart has been investigated using isoform-specific knockout (KO) mice. Systemic Nox2 homozygous KO, systemic Nox4 homozygous KO, and cardiac-specific Nox4 KO mice do not exhibit any obvious baseline cardiac phenotype, suggesting that Nox2 and Nox4 are mostly dispensable in the postnatal heart under baseline conditions [25,26,27]. Extracellular stresses, including G-protein coupled receptor agonists, mechanical stress, and hypoxia, however, activate Noxs in cardiomyocytes [28], suggesting that Nox isoforms are stress response proteins.

Nox isoforms are multi-transmembrane proteins (Figure 1). Nox1 forms a complex with p22*^phox^*, a transmembrane protein, and Noxo1, Noxa1, and Rac1, cytosolic proteins [29,30]. Nox1 generates ROS when it colocalizes with Noxo1 and Noxa1, homologs of p47*^phox^* and p67*^phox^*, respectively. Specifically, Noxa1 promotes Nox1 binding with Rac1, thereby promoting ROS generation [31]. Nox2 also forms a complex with p22*^phox^*, and its activity is regulated by interaction with cytosolic factors, such as p40*^phox^*, p47*^phox^*, p67*^phox^*, and Rac1 [32]. On the other hand, Nox4 is constitutively active, and its activity is regulated primarily by its expression level [33]. NF-κB is a critical regulator of Nox4 expression in cardiomyocytes in response to hypertrophic stimuli [34]. Nox2 and Nox4 protein levels are increased during I/R in the heart [15,35]. It should be noted that the activity of Nox4 can be regulated through post-translational mechanisms under some conditions. For example, polymerase δ-interacting protein 2 (POLDIP2) has been shown to positively regulate Nox4 activity by physically interacting with p22*^phox^* in VSMCs [36]. Fyn, a tyrosine kinase, negatively regulates Nox4 activity by phosphorylating Nox4 at Tyrosine 566 in the C-terminal domain, thereby interfering with the interaction between Nox4 and p22*^phox^* [37]. The functional significance of these post-translational modifications of Nox4 during myocardial I/R injury remains to be elucidated. Although Nox2 and Nox4 have similar structures, their regulation and subcellular localization appear to be distinct. Whereas Nox2 is localized primarily on the plasma membrane [38], Nox4 is located in the mitochondria, endoplasmic reticulum (ER), and nucleus [23,27,39]. Importantly, Noxs regulate a wide variety of cellular functions depending upon their subcellular localizations.

## 4. Detrimental Roles of Nox Isoforms-Derived ROS in Myocardial I/R Injury

### 4.1. Detrimental Roles of Nox Isoforms during Myocardial I/R Injury

The functions of endogenous Nox1, Nox2, and Nox4 in myocardial I/R injury have been investigated using Nox isoform-specific KO mice. Systemic Nox1, Nox2, and Nox1/Nox2 double KO mice were shown to exhibit a significant decrease in myocardial infarct size after I/R, but systemic Nox4 KO mice did not [14]. On the other hand, another study found that neither systemic Nox1 KO nor systemic Nox2 KO attenuated myocardial I/R injury in mice [35]. Our group has reported that both systemic Nox2 KO mice and cardiac-specific Nox4 KO mice exhibit a reduction in ROS production and infarct size after I/R [15]. Consistent with this study, downregulation of Nox4 by siRNA attenuated infarct size in mice after I/R [35]. Furthermore, cardiac-specific Nox4 transgenic mice exhibited increases in ROS production and infarct size in response to I/R [40]. These findings suggest that Nox2 and Nox4 play a detrimental role during myocardial I/R injury. To date, the roles of endogenous Nox3, Nox5, or Duox1/2 in myocardial I/R injury have not been reported.

### 4.2. Cell-Specific Roles of Nox Isoforms in Myocardial I/R Injury

It should be noted that there are some discrepancies among reports on the role of Nox4 in myocardial I/R injury. Although the loss of Nox4 function in cardiomyocytes exhibited protective effects, loss of Nox4 function in the germline did not [14,15,35,40]. One possible explanation for the difference could be that the function of Nox4 is cell-type specific. Although Nox4 overexpression in cardiomyocytes promotes apoptosis in vitro [23], Nox4 in endothelial cells exerts protective effects in lower limb ischemia by attenuating inflammation and enhancing angiogenesis [41]. These data indicate that Nox4 in cardiomyocytes may drive, whereas Nox4 in some other cell types may prevent, myocardial injury during I/R. The cardiac phenotypes of cardiac-specific Nox4 KO and systemic Nox4 KO were also observed to differ in terms of the effect of pressure overload: cardiac-specific Nox4 KO attenuates cardiac hypertrophy after transverse aortic constriction (TAC), whereas systemic Nox4 KO does not exhibit a protective effect [34]. Although the mechanism explaining the difference in the phenotype remains unknown, the cell type-specific roles of Nox4 may also explain the differences in cardiac phenotype between systemic and cardiac-specific Nox4 mice during pressure overload. 

### 4.3. ROS Production by Nox Isoforms during Myocardial I/R Injury

ROS induce oxidative damage in various organs when they exist in excess. We investigated the contributions of Noxs to the production of ROS during I/R. Systemic Nox2 KO and cardiac-specific Nox4 KO mice showed similar levels of reduction in ROS levels in the heart and in infarct size after I/R, suggesting that both Nox2 and Nox4 contribute to myocardial oxidative stress and damage in response to I/R, in spite of their distinct subcellular localizations [15]. In addition to oxidative damage of DNAs, lipids, and proteins, mitochondrial dysfunction, including mPTP opening, is intimately involved in I/R injury. Calcium-dependent mitochondrial swelling, which reflects mPTP opening, was decreased in systemic Nox2 KO and cardiac-specific Nox4 KO mice in response to I/R [15]. Netrin-1, a secreted axon guiding molecule, attenuated I/R-induced cardiac mitochondrial dysfunction via nitric oxide (NO)-dependent attenuation of Nox4 activation and recoupling of endothelial NO synthase (eNOS). Thus, the Nox4–uncoupled eNOS–mitochondrial dysfunction axis mediates myocardial I/R injury [35]. ER stress also critically mediates ROS-induced myocardial I/R injury. Expression of cleaved caspase-12, a mediator of ER stress-induced apoptosis, in the ischemic region was significantly increased by I/R in wild type (WT) mice but was reduced in both systemic Nox2 KO and cardiac-specific Nox4 KO mice [15].

## 5. Beneficial Aspects of Nox Isoforms in Myocardial I/R Injury

### 5.1. The Physiological Level of ROS Derived from Noxs during Myocardial I/R Injury

Whereas excessive ROS derived from Noxs mediate myocardial I/R injury, ROS also play a physiological role in the heart under stress conditions. For example, a small amount of ROS plays a protective role against ischemic injury by inducing preconditioning [42]. Nox2/Nox4 double KO (DKO) mice and transgenic mice with cardiac-specific overexpression of dominant negative (DN)-Nox, which broadly suppresses all Nox isoforms, exhibited larger infarcts than control mice in response to I/R [15]. Importantly, the level of ROS is below physiological levels in DKO and transgenic DN-Nox mice, in contrast to in single Nox KO mice, indicating that markedly reduced levels of ROS are detrimental during I/R [15]. Basal levels of ROS derived from either Nox2 or Nox4 are required for maintaining homeostasis in the heart during I/R. Thus, ROS play both physiological and pathological roles during myocardial I/R injury.

Energy metabolism is essential for maintaining cardiac function. Under basal conditions, the heart mainly uses fatty acids as a substrate to produce ATP. On the other hand, glycolytic genes are upregulated by hypoxia-inducible factor-1α (HIF-1α) during hypoxia in order to maintain ATP production and prevent increases in ROS production [43]. Furthermore, HIF-1α inhibits fatty acid oxidation (FAO) through suppression of peroxisome proliferator-activated receptor-α (PPARα), a master regulator of FAO, in the heart. Although protein levels of HIF-1α and PPARα were preserved in Nox2 or Nox4 single KO mouse hearts after I/R, HIF-1α was downregulated and PPARα upregulated in transgenic DN-Nox mouse hearts after I/R [15]. Genetic deletion of proline hydroxylase 2 (PHD2), an enzyme that induces hydroxylation and degradation of HIF-1α, normalized downregulation of HIF-1α and the increases in infarct size in transgenic DN-Nox mice after I/R. In addition, deletion of PPARα attenuated myocardial triglyceride accumulation and I/R injury in transgenic DN-Nox mice [15]. Thus, basal levels of ROS derived from either Nox2 or Nox4 suppress I/R injury through activation of HIF-1α and consequent suppression of PPARα (Figure 2).

### 5.2. Reductive Stress in Myocardial I/R Injury

Nicotinamide adenine dinucleotide (NAD^+^)/reduced NAD^+^ (NADH), phosphorylated NAD^+^ (NADP^+^)/reduced NADP^+^ (NADPH), and reduced glutathione (GSH)/GSH disulfide (GSSG) are critical redox couples that maintain the reducing environment in cells by serving as cofactors or substrates for neutralization of ROS. These redox couples are also intimately linked with cellular energetics. NADH is a pivotal electron donor during mitochondrial oxidative phosphorylation (OXPHOS), whereas NAD^+^ is an electron acceptor during glycolysis. NADPH is used as a major electron source for reductive biosynthesis of fatty acids and nucleic acids [44,45]. Disturbance of redox conditions provokes redox stress, including both oxidative stress and reductive stress. Compared to the wealth of knowledge regarding oxidative stress, reductive stress has been investigated less intensively. Reductive stress was first proposed in the context of hypoxia and reperfusion in hepatocytes, where reductive stress is stimulated when electron carriers are reduced under hypoxic conditions and re-oxidized after the oxygen supply is restored, leading to a burst of ROS generation [46]. Reductive stress has been generalized as a condition in which the balance between the cellular reducing capacity and pro-oxidant levels shifts to favor the reducing capacity, leading to excess accumulation of reducing equivalents (NADH, NADPH, and GSH) exceeding the capacity of endogenous oxidoreductases [47,48]. Broad suppression of Nox isoforms induces reductive stress during I/R injury. Overexpression of DN-Nox induces a markedly reduced state characterized by decreased NAD(P)^+^/NAD(P)H and increased GSH/GSSG ratios. Increased reductants paradoxically promote mitochondrial ROS production through the direct transfer of electrons to oxygen in the mitochondrial ETC during ischemia, resulting in no recovery of heart function after reperfusion (Figure 3). Since Nox activities are intimately associated with other oxidases in the ETC, optimal regulation of Nox-derived ROS levels is important to prevent both oxidative and reductive stress during myocardial I/R injury.

### 5.3. The Role of Nox4 in ER in Myocardial I/R Injury

Autophagy, a catabolic process in cells that delivers misfolded proteins and dysfunctional organelles to lysosomes for digestion, has been shown to exert beneficial effects during myocardial ischemia and reperfusion [49]. Oxidative stress can regulate autophagy either positively or negatively [50]. Nox4-derived ROS are intimately involved in autophagy regulation during glucose deprivation in cardiomyocytes [51,52]. Nox4 rapidly accumulates in the ER in response to energy stress and promotes autophagy through activation of the protein kinase RNA-like ER kinase (PERK) pathway by suppressing ER-specific prolyl hydroxylase 4 (PHD4). Reactivation of autophagy by Atg7 overexpression rescued a decrease in cell survival during glucose deprivation in the presence of Nox4 knockdown, suggesting that autophagy critically mediates the effect of Nox4 on cell survival. Nox4 is activated not only during fasting but also in the presence of prolonged ischemia in the heart, where Nox4 is also required for autophagy activation and cardioprotection (Figure 4). The effect of Nox4 upon autophagy is made possible by its ability to produce H_2_O_2_, a stable signaling molecule, at the ER. The functional role of Nox4 in autophagy regulation during myocardial reperfusion remains to be elucidated.

### 5.4. The Role of Nox4 at MAMs during Myocardial I/R Injury

Nox4 can be found at ER-mitochondria contact sites (MAMs) in multiple cell types and tissues, including cardiomyocytes and the heart [53]. I/R-induced necrosis of cardiomyocytes caused by mPTP opening is exacerbated in Nox4-null hearts compared to in WT hearts in the setting of the ex vivo Langendorff system. The beneficial effect of Nox4 was mediated by InsP_3_ receptor-dependent Ca^2+^ release into mitochondria at MAMs. These findings appear to contradict a previous study showing the detrimental roles of Nox4-derived ROS in I/R injury [15]. The discrepancy may be in part due to the difference in the model of I/R, namely in vivo versus ex vivo preparations, together with the use of cardiac-specific versus systemic deletion of Nox4. However, another possibility is that the function of Nox4 in the heart differs depending on its subcellular localization. For example, mitochondrial Nox4 exerts detrimental effects due to ROS-induced mitochondrial dysfunction during I/R, whereas Nox4 in ER and at MAMs may play protective roles against I/R injury through regulation of autophagy and MAM function, respectively (Figure 4). The Nox4 levels in each subcellular space may change differently during I/R. Thus, controlling the activity of Nox4 in a subcellular localization-specific manner may allow better control of I/R injury.

## 6. Novel Regulatory Mechanisms of Nox and Its Crosstalk with Other Oxidases

I/R injury regulates Noxs through multiple mechanisms. Cyclophilin A, a ROS-induced factor, attenuates hypoxia/reoxygenation-induced cardiomyocyte apoptosis via the AKT/Nox2 pathway [54]. A recent study demonstrated that miR-146a directly regulates Nox4 at the transcriptional level and protects against myocardial I/R injury [55]. Neuregulin-1 attenuates oxidative stress and inflammation by inhibiting Nox4 and NLR family pyrin domain containing 3 (NLRP3)/caspase-1 in myocardial I/R injury [56]. Klotho, an anti-aging single-pass membrane protein, decreases Nox2 and Nox4 levels in cardiomyocytes during I/R [57]. These upstream regulators of Nox2 and Nox4 are potential targets for the treatment of myocardial I/R injury.

Nox isoforms regulate the expression of one another, as well as other oxidases [58]. Knockdown of Nox4 upregulates mRNA and protein expression of Nox2, and knockdown of Nox2 increases the expression of Nox4 in endothelial cells [59]. Activation of Nox in endothelial cells induces endothelial nitric oxide synthase (NOS) uncoupling [60]. Conversely, NO production from coupled eNOS leads to NO-dependent inhibition of Nox4 [35]. Furthermore, Noxs are intimately involved in mitochondrial ETC-derived ROS production through the regulation of cellular NAD(P)^+^/NAD(P)H and increased GSH/GSSG levels [47,48]. The fact that inhibition of either Nox2 or Nox4 is effective despite activation of both isoforms during I/R injury might be explained by the fact that they crosstalk and enhance each other’s activities. Namely, both Nox2 and Nox4 contribute to ROS levels through positive feedback mechanisms during I/R [15]. Thus, interventions of either Nox2 or Nox4 alone may disrupt the feedforward mechanisms.

Mitochondria-derived ROS production precedes the activation of Nox during anoxia and reoxygenation. [61]. Complex I is a major source of mitochondrial superoxide during I/R [62]. Complex I produces superoxide during the forward electron transport, which occurs when the flavin mononucleotide (FMN) is reduced by accepting electrons from NADH [63]. Another mode of superoxide production by complex I is known as the reverse electron transport (RET), which occurs when electrons are forced to be transported backward through complex I [64]. Accumulation of succinate as an electron sink during ischemia drives a superoxide burst from complex I by RET during reperfusion [11]. Succinate accumulation is related to NADH metabolism [65]. Therefore, mitochondrial ROS production by RET may be linked to Nox activation.

## 7. Therapeutic Approach of Intervention into Nox Isoforms in I/R Injury

Growing evidence suggests that optimal regulation of Nox-derived ROS is an important therapeutic goal in the treatment of myocardial I/R injury. In particular, pharmacological approaches are better suited as strategies to reduce ROS levels more precisely compared to genetic manipulations such as genetic knockout of Nox2 and Nox4. GLX481304, a small molecule inhibitor of Nox2 and Nox4, improves contractile function by partially reducing Nox-derived ROS after I/R in the mouse heart [66]. A recent study demonstrated that petunidin, an anthocyanin, attenuates I/R injury in isolated hearts and anoxia/reoxygenation injury in cardiomyocytes by suppressing Nox4, but not Nox2, suggesting that Nox4 is a potential pharmacological target in I/R injury [67]. Metformin attenuates ROS production and myocardial I/R injury via adenosine 5′-monophosphate-activated protein kinase (AMPK)-dependent suppression of Nox4 [68]. We have recently demonstrated that teneligliptin, a dipeptidyl peptidase-IV (DPP-4) inhibitor, decreases Nox4 expression levels by increasing glucagon-like peptide-1 (GLP-1) in hearts treated with angiotensin II [69]. Furthermore, inhibition of DPP-4 has been shown to attenuate myocardial I/R injury by enhancing the endogenous GLP-1 level [70]. These findings suggest that a DPP-4 inhibitor/GLP-1 agonist could effectively regulate Nox4 in the heart during I/R.

A clinical trial of GKT137831 (also called setanaxib), the first-in-class Nox1/4 inhibitor, in patients with type 2 diabetes has been reported [Clinicaltrials.govreferenceNCT02010242]. Although GKT137831 failed to improve the primary endpoint of albuminuria, it exhibited salutary effects on the secondary endpoints of inflammation and liver dysfunction and showed an excellent safety profile. Another clinical trial in patients with type 1 diabetes is now ongoing [71]. This class of Nox inhibitors may be applied for the treatment of I/R injury in the future.

## 8. Future Perspectives

Several questions remain unanswered regarding Noxs in myocardial I/R injury. First, the regulatory mechanism and the function of Nox4 in various subcellular localizations and organelles remain to be clarified. Since the diverse roles of Nox4 may depend on its subcellular localization, it may be necessary to consider location-specific regulation of Nox4 for the treatment of I/R injury. Second, it is important to determine the optimal levels of ROS to avoid both oxidative and reductive stress. Third, the crosstalk among Noxs, other oxidases, and mitochondrial ROS production, including the succinate-related RET, needs to be elucidated. Finally, therapies targeting Noxs with small molecules have not yet been established for myocardial injury in the clinical setting. The development of safe, specific, and effective Nox inhibitors is urgently needed.

## 9. Conclusions

In this review, we discussed the functional significance of ROS derived from Noxs in myocardial I/R injury. Both Nox2 and Nox4 are involved in the pathophysiology of I/R injury in a complex manner. Whereas Nox4 on the mitochondrial membrane and Nox2 on the plasma membrane may both facilitate oxidative stress through feedforward mechanisms, Nox4 on the ER membrane or at MAMs contributes to salutary effects through induction of autophagy and improvement of Ca^2+^ homeostasis. Several upstream regulators and inhibitory drugs of Nox2 and Nox4 have been identified. Although Noxs are potential therapeutic targets for I/R injury, achieving the optimal levels of ROS for the maintenance of physiological cellular functions appears essential in order to maximize the effectiveness of interventions. In addition, selective modulation of Nox4 at specific subcellular locations also appears important.

## Figures and Tables

**Figure 1 antioxidants-11-01069-f001:**
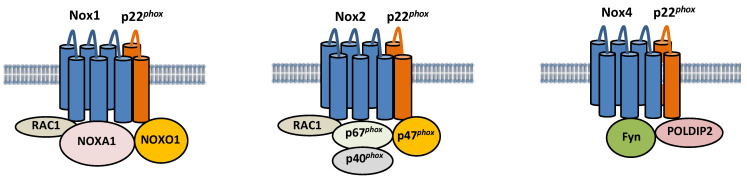
Composition of Nox1, Nox2, and Nox4. Nox1, 2, and 4 are six-transmembrane proteins. Nox1 forms a complex with p22*^phox^*, Noxo1, Noxa1, and Rac, and its activity is regulated by the interaction. The activity of Nox2 is also regulated by interaction with cytosolic factors, such as p40*^phox^*, p47*^phox^*, p67*^phox^*, and Rac1. The activity of Nox4 is positively and negatively regulated by POLDIP2 and Fyn, respectively.

**Figure 2 antioxidants-11-01069-f002:**
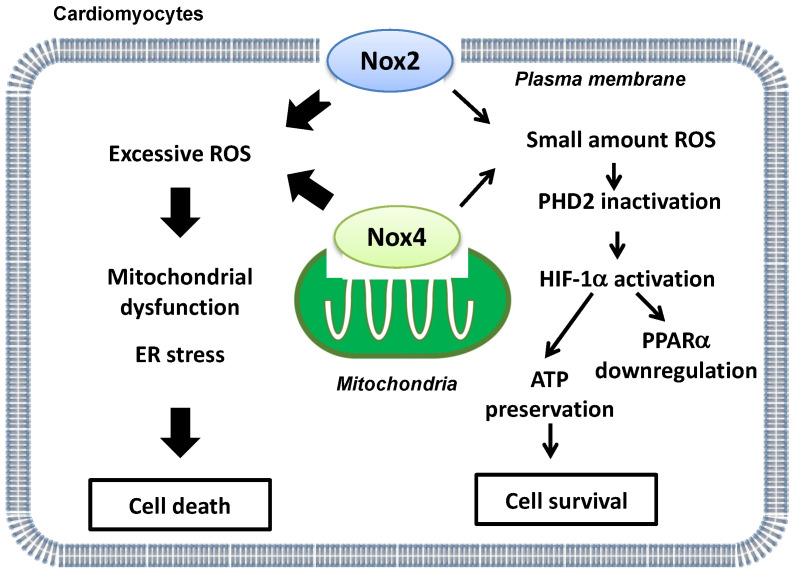
A schematic representation of ‘Yin and Yang’ aspects of Nox2 and Nox4-derived ROS during myocardial I/R injury. Whereas excessive ROS production derived from Nox2 and Nox4 is detrimental (**left side**), basal ROS levels derived from either Nox2 or Nox4 are indispensable for energy metabolism and ATP production during I/R (**right side**).

**Figure 3 antioxidants-11-01069-f003:**
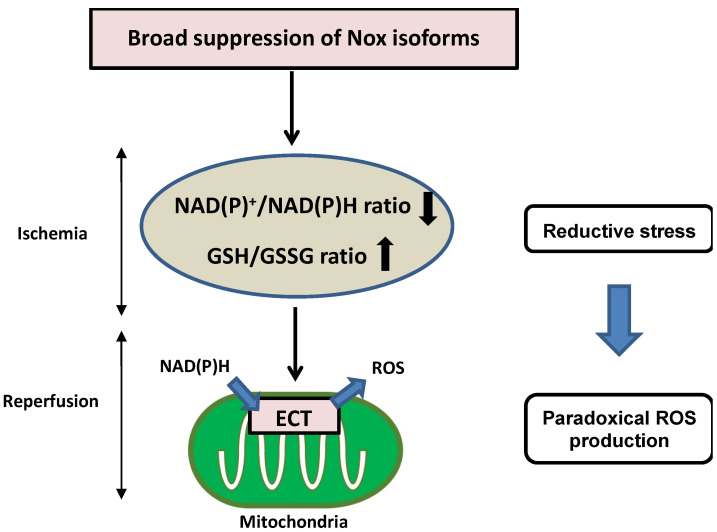
Detrimental effects of reductive stress during myocardial I/R injury. Broad suppression of Nox activity induces a markedly reduced state characterized by decreased NAD(P)^+^/NAD(P)H and increased GSH/GSSG. Increased reductants paradoxically promote mitochondrial ROS production through the direct transfer of electrons to oxygen in the mitochondrial ETC during ischemia, resulting in no recovery of heart function after reperfusion.

**Figure 4 antioxidants-11-01069-f004:**
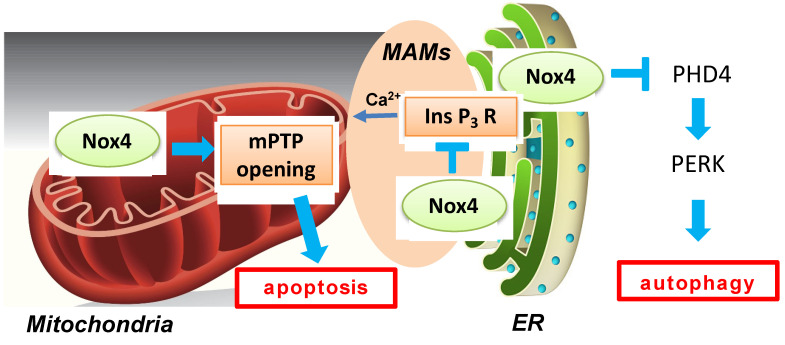
A schematic representation of multiple functions of Nox4-derived ROS during myocardial I/R injury. Mitochondrial Nox4 exerts a detrimental effect through ROS-induced mitochondrial dysfunction during I/R, whereas Nox4 on the ER membrane and at MAMs plays protective roles against I/R injury through regulation of autophagy and MAM function, respectively. Nox4 activates PERK by inhibiting PHD4 in ER. In addition, Nox4 suppresses calcium transfer from ER to mitochondria by inhibiting the Ins P_3_ receptor.

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
