# Peer review of "Yin and Yang of NADPH Oxidases in Myocardial Ischemia-Reperfusion"

_antioxidants, 2022, doi:10.3390/antiox11061069_

Round 1
Reviewer 1 Report
Review summarizes data physiological and pathological functions of NADPH oxidases (Nox2 and Nox4) in the heart and during myocardial I/R injury,
From this point is the review focused on an actual and important topic. Review is well written and organized.
I have only a few comments:
1. In lines 85-86 you wrote: “Nox1 forms a complex with p22phox, Noxo1, Noxa1, and Rac, and its activity is regulated by the interaction.”
The interaction is not clearly defined.
2. In lines 170-172 you wrote: “Importantly, levels of oxidative stress were significantly lower in DKO or transgenic DN-Nox mice than in single Nox KO mice, indicating that markedly reduced levels of ROS are detrimental during I/R.”
During oxidative stress is increased production of ROS. What does it mean, significantly lower level of oxidative stress? Are at these conditions levels of ROS markedly reduced (under basal physiological levels) and this is associated with suppression of their signaling role?
Author Response
We greatly thank the reviewer for the helpful and constructive comments toward improving our manuscript. We hope that the revised manuscript can now be considered for publication.
Comment 1: In lines 85-86 you wrote: “Nox1 forms a complex with p22phox, Noxo1, Noxa1, and Rac, and its activity is regulated by the interaction.” The interaction is not clearly defined.
Response:
Thank you for your comment. Nox1 generates ROS when it colocalizes with Noxo1 and Noxa1, homologues of p47phox and p67phox, respectively. Noxa1 promotes Nox1 binding with Rac1, thereby promoting ROS generation (Chen et al JBC 281, 17718-17726, 2006). We now include this explanation in our revised manuscript.
Lines 85-89
Nox1 forms a complex with p22phox, a transmembrane protein, and Noxo1, Noxa1, and Rac, cytosolic proteins. Nox1 generates ROS when it colocalizes with Noxo1 and Noxa1, homologues of p47phox and p67phox, respectively. Specifically, Noxa1 promotes Nox1 binding with Rac1, thereby promoting ROS generation (Chen et al JBC 281, 17718-17726, 2006).
Comment 2: In lines 170-172 you wrote: “Importantly, levels of oxidative stress were significantly lower in DKO or transgenic DN-Nox mice than in single Nox KO mice, indicating that markedly reduced levels of ROS are detrimental during I/R.” During oxidative stress is increased production of ROS. What does it mean, significantly lower level of oxidative stress? Are at these conditions levels of ROS markedly reduced (under basal physiological levels) and this is associated with suppression of their signaling role?
Response:
Thank you for raising an important point. The significantly lower levels of oxidative stress indicate that the level of reactive oxygen species is below physiological levels in these models. Under these conditions, inadvertent downregulation of hypoxia-inducible factor-1α and upregulation of peroxisome proliferator-activated receptor-α exacerbate I/R injury, as shown in Figure 2. We rephrased the paragraph as follows: ”Importantly, levels of oxidative stress were dramatically reduced in DKO or transgenic DN-Nox mice, in contrast to in single Nox KO mice”.
Lines 170-172
Importantly, levels of oxidative stress were dramatically reduced in DKO or transgenic DN-Nox mice, in contrast to in single Nox KO mice, indicating that …
Lines 188-190
Thus, basal levels of ROS derived from either Nox2 or Nox4 suppress I/R injury through activation of HIF-1α and consequent suppression of PPARα (Figure 2).
Reviewer 2 Report
The article “Yin and Yang of NADPH Oxidases in Myocardial Ischemia-reperfusion” has been reviewed. Generally speaking, this review contains necessary information about the topic which is basically pros and cons of NOx in IR injury. Supplementary information about NOx such as regulatory mechanisms of NOx and its interactions with other oxidases and treatment of IR injury is also included which makes the review a comprehensive one. In addition to textual depiction, the review applies figures to outline some biological pathways which is easy for people who do not familiar with the field to read and understand. The authors provided details about the prospects of the NOx related application and questions remain unsolved which shows that the authors know what to do next and their profession in the field.
However, I still have some advices:
First of all, the review could contain more information about the current research that people are conducting on NOx such as some actual experiments and the conclusions of those experiments. Secondly, the figure 2 is supposed to change position to the place after 5.1 because the context in the figure has not been completely discussed in the main text. The figure contains Yin and Yang of NOx but the text up here just illustrates the Yin of NOx which makes the figure unreasonable.
Author Response
We greatly thank the reviewer for the helpful and constructive comments toward improving our manuscript. We hope that the revised manuscript can now be considered for publication.
Comment 1: First of all, the review could contain more information about the current research that people are conducting on NOx such as some actual experiments and the conclusions of those experiments.
Response:
Thank you very much for this important suggestion. We made sure that we cite and discuss most of the recent papers reporting the relevant functions of Noxs in the heart. We also included a paper showing that metformin attenuates myocardial I/R injury through AMPK-dependent suppression of Nox4 (Shi and Hou, Mol Med Rep, 24, 712, 2021).
Lines 308-310
Metformin attenuates ROS production and myocardial I/R injury via adenosine 5'‑monophosphate‑activated protein kinase (AMPK)-dependent suppression of Nox4 (Shi and Hou, Mol Med Rep, 24, 712, 2021).
Comment 2: Secondly, the figure 2 is supposed to change position to the place after 5.1 because the context in the figure has not been completely discussed in the main text. The figure contains Yin and Yang of NOx but the text up here just illustrates the Yin of NOx which makes the figure unreasonable.
Response:
Thank you very much for this comment. We agree with the reviewer. We have now moved the position of “Figure 2” to after Line 204.
Round 2
Reviewer 2 Report
The revised article “Yin and Yang of NADPH Oxidases in Myocardial Ischemia-reperfusion” has been reviewed. In this revised version, the questions mentioned by reviewers before have been resolved carefully. After revision, the article was more readable, and I considered that the article was acceptable now.